# Creating tunable lateral optical forces through multipolar interplay in single nanowires

Fan Nan [1] ✉, Francisco J. Rodríguez-Fortuño [2], Shaohui Yan [3] ✉, Jack J. Kingsley-Smith [2], Jack Ng [4], Baoli Yao[3], Zijie Yan[5,6] & Xiaohao Xu [3] ✉

The concept of lateral optical force (LOF) is of general interest in optical manipulation as it releases the constraint of intensity gradient in tightly focused light, yet such a force is normally limited to exotic materials and/or complex light fields. Here, we report a general and controllable LOF in a nonchiral elongated nanoparticle illuminated by an obliquely incident plane wave. Through computational analysis, we reveal that the sign and magnitude of LOF can be tuned by multiple parameters of the particle (aspect ratio, material) and light (incident angle, direction of linear polarization, wavelength). The underlying physics is attributed to the multipolar interplay in the particle, leading to a reduction in symmetry. Direct experimental evidence of switchable LOF is captured by polarization-angle-controlled manipulation of single Ag nanowires using holographic optical tweezers. This work provides a minimalist paradigm to achieve interface-free LOF for optomechanical applications, such as optical sorting and light-driven micro/nanomotors.

The prevailing model of optical forces on single microscopic particles, which relies on the electric dipole approximation, assumes that the force experienced by a particle is influenced by the nonuniformity of the illumination field[1]. In this scenario, the particle experiences gradient forces due to the intensity gradient in a tightly focused optical field, or the phase gradient in an extended optical field where the varied phase redirects the radiation pressure[2-4]. Lateral optical force (LOF), acting along a direction with neither intensity nor phase gradient, is generally prohibited in this context[5-22]. LOF allows optical manipulation in a spatially extended light field (thus, laser heating can be circumvented), which is beyond the capabilities of optical tweezers in conventional wisdom[23]. Aside from laterally asymmetric objects (e.g., a single cambered rod)[5], LOF was discovered in single chiral objects with linear polarization, where an interface (e.g., reflective substrate) is employed to interact with the particle or yield evanescent fields possessing transverse spin[6,7,9]. Moreover, LOF can be induced in a nonchiral particle by spin–orbit coupling[8] and the design of near-field directionalities[24]. With more flexibility in practical optical manipulation, interface-free LOF has been an active pursuit of research. At the single-particle level, interface-free LOF has been theoretically proposed by using complex structured light[11,21]. However, very limited experimental evidence is provided. Only recently, Shi et al. reported a stable interface-free LOF caused by the inhomogeneities of the spin angular momentum[25]. In addition, there is no demonstration of interface-free LOF applied on a nanoparticle under the illumination of a single linearly polarized beam.

In complex multiparticle systems, phased nanoantenna arrays have provided insights for interface-free LOF based on electric dipole-dipole interaction, where asymmetric far-field radiation is easily achieved by tuning the orientation and retardation phase of the dipoles, as shown in Fig. 1a. For example, free-standing asymmetric nanodimers and their arrays can generate translational and rotational

[1]Guangdong Provincial Key Laboratory of Nanophotonics Manipulation, Institute of Nanophotonics, Jinan University, 511443 Guangzhou, China. [2]Department of Physics and London Centre for Nanotechnology, King's College London, Strand, London WC2R 2LS, United Kingdom. [3]State Key Laboratory of Transient Optics and Photonics, Xi'an Institute of Optics and Precision Mechanics, Chinese Academy of Sciences, 710119 Xi'an, China. [4]Department of Physics, Southern University of Science and Technology, 518055 Shenzhen, Guangdong, China. [5]Department of Applied Physical Sciences, University of North Carolina at Chapel Hill, Chapel Hill, North Carolina 27599, USA. [6]Deceased: Zijie Yan. ✉e-mail: fnan190730@gmail.com; shaohuiyan@opt.ac.cn; xuxhao_dakuren@163.com

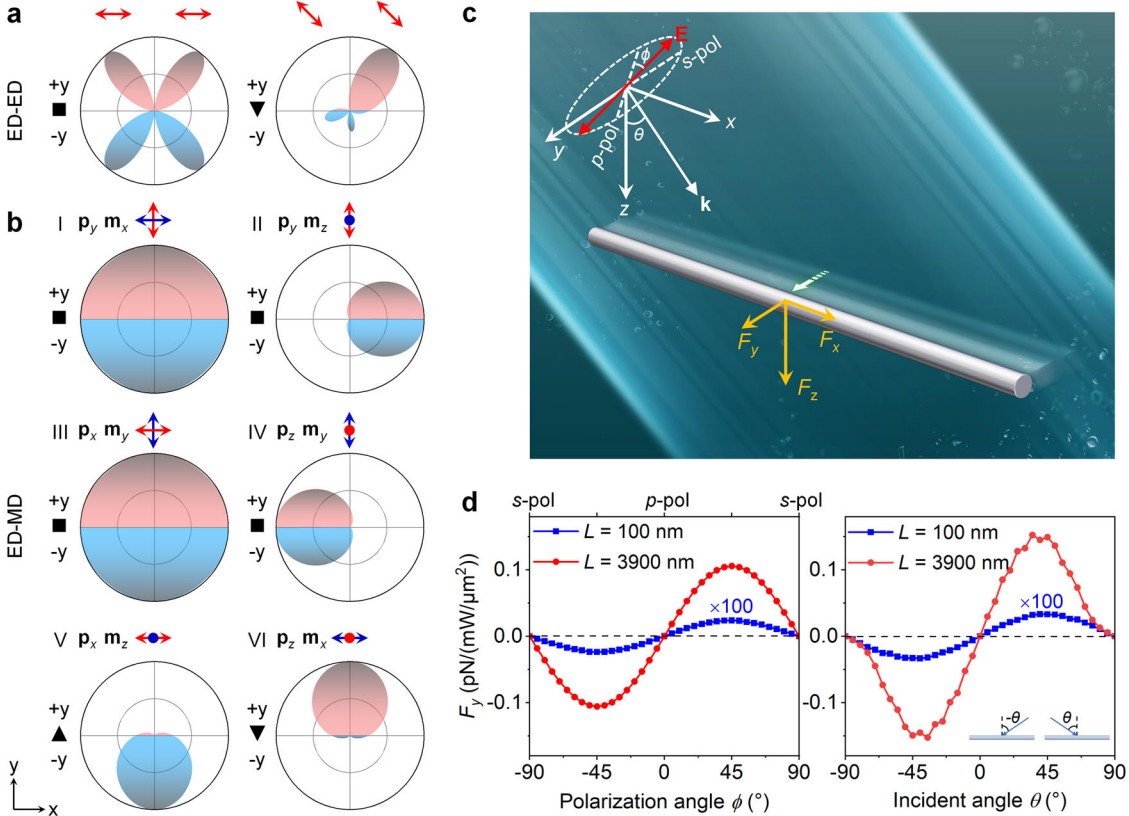

**Fig. 1 | Tunable LOF applied on a nonchiral elongated particle with a linearly polarized plane wave. a, b** Radiation patterns in the $x$–$y$ plane from (**a**) two parallel aligned electric dipoles (ED) with different excitation phases and polarization angles. The retardation phases in the left and right panels are $\pi$ and $\pi/2$, respectively. They can be introduced by controlling the incident angle of a plane wave. The separation of two dipoles is fixed at $0.5\lambda$ and their orientations are indicated by the red arrows. This is a general working principle to create LOF (in the $x$–$y$ plane) by assembled nanoparticles but only if they could be simplified to an electric dipole array with specific phase retardations. **b** The proposed LOF system based on ED and magnetic dipole (MD) interaction within a small nanoparticle. The ED and MD are placed in the $x$–$y$ plane and their orientations are indicated by the red and blue arrows, respectively. Here the magnitudes of ED and MD satisfy $|\mathbf{p}|^2 = |\mathbf{m}/c|^2$, which is known as Kerker's condition. For cases of V and VI, the electric-magnetic symmetry is preserved along $y$ direction because $p_x^*/p_z^* = m_x/m_z$. The red and blue colors indicate optical scattering along $+y$ and $-y$ directions, respectively. **c** Schematic of the illumination of a free-standing Ag cylinder by a linearly polarized plane wave in water. The orientation of the cylinder is along the $x$ axis. The lateral force can emerge in the $y$ direction which relies on the polarization and incident angles ($\phi$, $\theta$) of the optical field. **d** Left: Numerically calculated LOF ($F_y$, total optical force along the $y$ axis) of a short (100 nm) and a long (3900 nm) Ag cylinder as a function of the polarization angle of an oblique plane wave ($\lambda = 800$ nm, $\theta = 20°$). The diameter ($d$) of Ag cylinder is fixed at 80 nm. For both lengths, variations of $F_y$ with respect to $\phi$ are well-fitted by a sinusoidal dependence. Right: Numerically calculated LOF of the two Ag cylinders as a function of the incident angle of the plane wave ($\lambda = 800$ nm, $\phi = 45°$).

motions by directional scattering in plane waves[17,26–28]. However, these free-standing micro/nanostructures usually suffer from complex fabrication with low throughput, which largely hinder the application of LOF in light-driven micro/nanomotors[29–31].

Here, we present a previously unidentified paradigm to create tunable LOF by exploiting multipolar interplay in a single elongated particle (e.g., Ag nanowire), which has a symmetric lateral geometry. More importantly, this LOF is a pure recoil effect, whose excitation requires neither transverse momentum nor transverse spin. Through computational analysis, we find that when the nanowire is short (i.e., can be described by dipoles), the interference between electric dipole and magnetic dipole is the main contribution to the LOF (Fig. 1b). For a longer nanowire, the dominant contributor will switch to the interference between higher multipoles such that the LOF applies to nanowires with various lengths. Consequently, this LOF is sign-reversible and magnitude-tunable, which is feasible by modulating the direction of linear polarization, incident angle, and wavelength of a simple plane wave. It also depends on the aspect ratio and material of the nanowire. Experimentally, we have achieved on-demand transportation and bidirectional releasing of a single Ag nanowire by polarization angle-controlled LOF. These results offer useful insights

for optical manipulation by reconfigurable LOF, which may be useful for optomechanical and biological applications.

## Results

### Interface-free LOF arising from asymmetric light scattering with linear polarization

We approximate the Ag nanowire as a cylinder with variable lengths, whose long axis aligns to the $x$ axis, as shown in Fig. 1c. The plane wave is assumed to be linearly polarized, with propagation vector **k** in the $x$–$z$ plane, so that its electric and magnetic vectors can be written as

$$\mathbf{E} = (-\cos\theta \cos\phi \, \mathbf{e}_x + \sin\phi \, \mathbf{e}_y + \sin\theta \cos\phi \, \mathbf{e}_z) \exp[i(k_x x + k_z z)]$$
$$\mathbf{B} = \frac{1}{c}(-\cos\theta \sin\phi \, \mathbf{e}_x - \cos\phi \, \mathbf{e}_y + \sin\theta \sin\phi \, \mathbf{e}_z) \exp[i(k_x x + k_z z)],$$
$$(1)$$

where we have assumed the time dependence $\exp(-i\omega t)$; $\mathbf{e}_x$, $\mathbf{e}_y$, and $\mathbf{e}_z$ are the unit vectors in Cartesian coordinates; the incident angle (with respect to the $z$ direction) and the polarization angle (with respect to the $p$-polarization) is defined as $\theta$ and $\phi$, respectively. Our full-wave numerical simulations clearly prove the existence of an extraordinary LOF (Fig. 1d), although the cylinder is completely symmetric about the

$x$–$z$ plane. It shows that a deviation from normal incidence ($z$- and $x$-axes) leads to the LOF, which vanishes at $s$- and $p$- polarizations. Moreover, both the sign and magnitude of the lateral force can be tuned by modulating the linear polarization angle and angle of incidence. Specifically, the cylinders experience a force maximum when the polarization angle is $\pm 45°$.

To understand the origin of the LOF, we begin for simplicity with the dipole model of the optical force[32]:

$$\mathbf{F} = \mathbf{F}^e + \mathbf{F}^m + \mathbf{F}^{em},$$
$$\mathbf{F}^e = \tfrac{1}{2}\mathrm{Re}[(\nabla\mathbf{E}^*)\cdot\mathbf{p}], \ \mathbf{F}^m = \tfrac{1}{2}\mathrm{Re}[(\nabla\mathbf{B}^*)\cdot\mathbf{m}], \ \mathbf{F}^{em} = -\tfrac{k^4}{12\pi\varepsilon c}\mathrm{Re}(\mathbf{p}^*\times\mathbf{m}), \quad (2)$$

where $c$ is the speed of light in the ambient medium with permittivity $\varepsilon$ and permeability $\mu$. The electric and magnetic dipole moments can be expressed as

$$\mathbf{p} = \overset{\leftrightarrow}{\alpha}^e\mathbf{E}, \ \mathbf{m} = \overset{\leftrightarrow}{\alpha}^m\mathbf{B}, \quad (3)$$

where

$$\overset{\leftrightarrow}{\alpha}^e = \begin{bmatrix} \alpha_\parallel^e & 0 & 0 \\ 0 & \alpha_\perp^e & 0 \\ 0 & 0 & \alpha_\perp^e \end{bmatrix}, \ \overset{\leftrightarrow}{\alpha}^m = \begin{bmatrix} \alpha_\parallel^m & 0 & 0 \\ 0 & \alpha_\perp^m & 0 \\ 0 & 0 & \alpha_\perp^m \end{bmatrix} \quad (4)$$

are the electric and magnetic polarizability tensors. Note that we use SI units throughout the paper. It is readily to verify that $\mathbf{F}^e$ and $\mathbf{F}^m$, which are the forces due to light extinction, always lie in the $x$–$z$ plane for our illumination geometry.

The last term in Eq. (2), $\mathbf{F}^{em}$, represents a recoil force due to asymmetric far-field light scattering[32]. This asymmetric scattering from multipolar interference is the same force used in tractor beams[33,34] and, for dipolar scatterers, corresponds to the recoil force of a "Huygens source"[35–37] which radiates mostly in the $\mathrm{Re}(\mathbf{p}^*\times\mathbf{m})$ direction. Therefore, the LOF will be induced as long as the interference term $\mathrm{Re}(\mathbf{p}^*\times\mathbf{m})$ possesses the lateral component, which is given by $\mathrm{Re}(p_z^*m_x - p_x^*m_z)$. It is evident that the LOF is zero at $s$-polarization (where $p_x = p_z = 0$) and $p$-polarization ($m_x = m_z = 0$); the vanishing LOF can also be understood by the radiation symmetry, as exemplified in Fig. 1b-I, II and Fig. 1b-III, IV for $s$- and $p$-polarization, respectively. Only when $p_z$ and $m_x$ (or $p_x$ and $m_z$) coexist could the LOF be produced (Fig. 1b-V, VI), but it also requires the breaking of electric-magnetic symmetry, i.e., $p_x^*/p_z^* \neq m_x/m_z$. These essential prerequisites are easily satisfied in an elongated nanoparticle (e.g., a nanocylinder), whose longitudinal and transverse modes can be excited simultaneously under the illumination of an obliquely incident plane wave. In fact, substituting Eqs. (1) and (3) into $\mathbf{F}^{em}$, one may express its lateral component as

$$F_y^{em} = \frac{\mu k^4}{48\pi}\sin(2\theta)\sin(2\phi)\,\mathrm{Re}(\alpha_\perp^{e*}\alpha_\parallel^m - \alpha_\parallel^{e*}\alpha_\perp^m), \quad (5)$$

which always vanishes for spheres ($\alpha_\perp^e = \alpha_\parallel^e, \alpha_\perp^m = \alpha_\parallel^m$) as it should, but is generally nonzero for the nanocylinders that have anisotropic polarizabilities. The LOF in our numerical simulations (Fig. 1d) can be well explained by Eq. (5) for the short Ag cylinder (i.e., $L = 100$ nm, $d = 80$ nm); its sinusoidal-like dependence on both the polarization and incident angle is attributed to the pre-factors, $\sin(2\theta)$ and $\sin(2\phi)$, in the formula (see Supplementary Fig. 1a, b for more details). These theoretical results also suggest that the LOF stems from the recoiling effects for dipoles, which represent the lowest-order multipoles.

With the increase of the cylinder length (e.g., $L = 3900$ nm), higher-order multipoles may exist in the cylinder, so the dipole model will no longer hold exactly. Nevertheless, the dipole model still provides a good qualitative description of the LOF (Fig. 1d, also see Supplementary Fig. 1c). We have checked the recoil forces applied on a series of Ag cylinders with different lengths and diameters, which are

identical with the LOF applied on the same cylinder (Supplementary Fig. 2). These numerical results clearly verify that the induced LOF is a pure recoil effect in our system. Using the method of Cartesian multipole expansion, one may write the recoil force related to all possible multipoles as[22,38]

$$\mathbf{F}_{rec} = \sum_{l=1}^{\infty}\left[\mathbf{F}^{x(l)} + \mathbf{F}^{e(l)} + \mathbf{F}^{m(l)}\right] \quad (6)$$

$$\mathbf{F}^{x(l)} = \frac{1}{4\pi\varepsilon c}\frac{k^{2l+2}}{l\,l!(2l+1)!!}\mathrm{Re}\left[\mathbb{O}_{elec}^{(l)}(l-1)\cdot\cdot\mathbb{O}_{mag}^{(l)*}(2)\cdot\cdot\overset{\leftrightarrow}{\overset{\leftrightarrow}{\epsilon}}\right] \quad (7)$$

$$\mathbf{F}^{e(l)} = -\frac{1}{4\pi\varepsilon}\frac{(l+2)k^{2l+3}}{(l+1)!(2l+3)!!}\mathrm{Im}\left[\mathbb{O}_{elec}^{(l)*}(l)\cdot\cdot\mathbb{O}_{elec}^{(l+1)}\right] \quad (8)$$

$$\mathbf{F}^{m(l)} = -\frac{\mu}{4\pi}\frac{(l+2)k^{2l+3}}{(l+1)!(2l+3)!!}\mathrm{Im}\left[\mathbb{O}_{mag}^{(l)*}(l)\cdot\cdot\mathbb{O}_{mag}^{(l+1)}\right]. \quad (9)$$

where $\overset{\leftrightarrow}{\overset{\leftrightarrow}{\epsilon}}$ is the Levi-Civita tensor, $\mathbb{O}_{elec}^{(l)}$ denotes the electric $2^l$-pole moment, and $\mathbb{O}_{mag}^{(l)}$ denote the magnetic $2^l$-pole moment; for $l=1$, $\mathbb{O}_{elec}^{(l)} = \mathbf{p}$ and $\mathbb{O}_{mag}^{(l)} = \mathbf{m}$ so that Eq. (7) reduces to $\mathbf{F}^{em}$. Equations (6)–(9) indicate that the recoil force is not limited to the dipoles, but is also found in higher multipoles, including their hybrid magnetoelectric interaction (Eq. (7)) and the interaction between purely electric (or magnetic) modes (Eqs. (8) and (9)). These multipolar recoil forces can have a lateral component in our system. We shall show below that the higher multipoles indeed contribute to the LOFs on longer nanocylinders by observing light scattering behaviors.

## Reconfigurable and switchable LOF

To explore the multipole effects on the LOF, we first checked the polarization angle-resolved scattering cross section and LOF, for nanowires with different lengths (Supplementary Fig. 3). We find that the LOF on each cylinder reaches a maximum for the polarization angle $\phi = \pm 45°$. Then we calculate the scattering spectrum as a function of $L$ and $\theta$, as shown in Fig. 2a, where $\phi$ is fixed at 45°. The coupling of light with the Fabry-Perot (FP) and Mie resonances lead to multipole modes, which can be effectively tuned by the incident angle. The LOFs have experienced abrupt changes due to the emergence of the multipoles (Fig. 2b). In the range of 0–20°, switchable LOF is achieved as the cylinder length changes. Moreover, the magnitude of LOF oscillates in longer cylinders (also see Supplementary Fig. 4). The sign of LOF shows multiple reversals with the increment of cylinder length at a small angle of incidence (e.g., $\theta = 2°$). However, when $\theta$ increases to 20°, the LOF is predominantly positive, which increases significantly with cylinder length.

We calculate the far-field scattering patterns of different Ag cylinders in the $x$–$y$ plane. At normal incidence ($\theta = 0°$ or 90°), the cylinders scatter the same amount of photons in opposing directions for all polarization angles, thus the cylinder does not experience any LOF. For example, Supplementary Fig. 5 shows the far-field scattering patterns of a Ag cylinder for polarization angles $\phi = 0°$, −45°, 45°, and 90°. At oblique incidence, the symmetry of far-field light scattering is broken when $\phi \neq 0°$ and 90°, and directional scattering takes opposite sign by switching the sign of $\phi$. Figure 2c, d shows a series of far-field scattering patterns of five selected Ag cylinders with $\theta = 2°$ and 20°, respectively. With a short length (e.g., $L = 100$ nm), the scattering patterns show a normal dumbbell shape (i.e., dominated by electric dipolar radiation). As the length increases, the interference of higher-order multipoles gives rise to more complex radiation patterns. For a length of 350 nm, the pattern retains its two-lobe symmetry with $\theta = 2°$. However, it transforms into four petals with $\theta = 20°$. When $L = 1500$ nm, multi-petal patterns with two major lobes are observed.

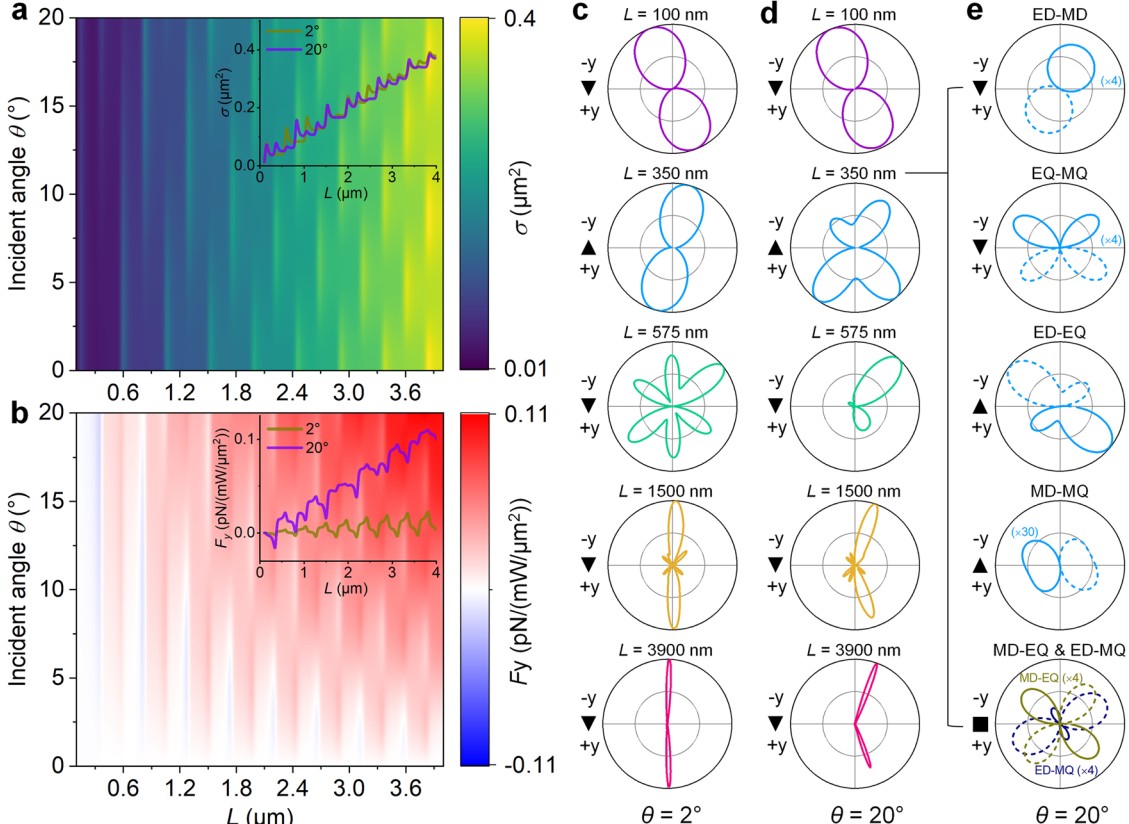

**Fig. 2 | Tuning LOF by cylinder length. a** Calculated incident angle-resolved scattering cross section of a Ag cylinder with different length $L$. The abrupt changes shown in the scattering spectrum are caused by the optical resonance. The inset shows the scattering cross sections at two representative angles. **b** Calculated LOFs on the Ag cylinder with different lengths. The inset shows the LOFs at two representative angles. **c**, **d** Normalized far-field angular distributions of the scattered intensity of selected Ag cylinders with **c** small (2°) and **d** large angles (20°) of incidence. **e** The absolute values of the radiation intensity distribution resulting from the interaction between different multipoles calculated for $L$ = 350 nm. Dashed and solid lines represent the negative and positive contributions, respectively, due to the destructive and constructive interference. For all calculations, the cylinder diameter and the polarization angle are fixed at 80 nm and 45°, respectively.

The two major lobes become narrower and directional radiation is stronger as the length further increases to 3900 nm.

To see whether higher multipoles are involved in the asymmetry radiation and hence the LOF, we perform a mode expansion analysis with multipole theory[39–43]. Because the radiation of single multipoles is always symmetric[41], the asymmetric scattering must originate from the interaction between different multipoles. Therefore, we focus on the radiation associated with the interference terms of the scattering coefficients with different order $l$ or type (electric/magnetic) (also see "Methods"). As a representative demonstration, Fig. 2e shows the radiation patterns created by different interacting modes in the Ag cylinder with $L$ = 350 nm. In contrast to the total scattering intensity which is always positive (Fig. 2d), these interplay-related components can be negative in some directions (indicated by dashed lines), suppressing the net scattering due to destructive interference, as explained in "Methods".

For the ED-MD interference (the first row), the scattering intensity is mainly negative (or destructive) in the lower half arc, while being positive (or constructive) in the upper. From such asymmetric scattering, one can expect the emergence of a recoil force in the lateral direction, which has already been captured by our dipole model (Eq. (5)). Remarkably, the constructive and destructive interference also occur when the electric quadrupole (EQ) interacts with the magnetic quadrupole (MQ) or ED, resulting in unbalanced radiation from which the LOF can be expected. The same is true for the MD-MQ interplay, whose scattering intensity, despite small, exhibits non-symmetric angular dependence. However, the interplay between ED-MQ or EQ-

MD results in a symmetric profile, which is not responsible for the LOF. This is in agreement with Eqs. (7)–(9), and the Lorenz–Mie theory[44], that the recoiling optical force on a single particle can only stem from the interaction between hybrid (magnetoelectric) modes with the same order, or purely electric (or magnetic) modes of adjacent orders. It is obvious that the ED-EQ interplay is the main contributor to the LOF for this cylinder. A similar LOF mechanism can be expanded beyond the quadrupoles. For example, asymmetric radiation profiles can be produced by the octupoles excited in longer nanocylinders (e.g., $L$ = 575 nm, Supplementary Fig. 6).

This LOF is also sensitive to the diameter of the cylinder. Figure 3a shows that a Ag cylinder with smaller diameter (e.g., $d$ = 80 nm) only support a certain direction of optical forces ($F_y > 0$), whose magnitude is increased with the incident angle (in the range of 0–20°). However, the sign of LOF experiences multiple reversals with a diameter of 140 nm. As the diameter increases to 300 nm, the LOF undergoes a stronger oscillation whose sign is negative for all incident angles (in the range of 0–20°). Interestingly, the oscillation amplitude and phase can both be tuned by the diameter (see the blue and olive curves). Furthermore, switchable LOF is feasible by tuning the wavelength of the plane wave (Fig. 3b). These results can be understood as the changes of the size of the cylinder (with a fixed wavelength) and the excitation wavelength of the light (with a fixed cylinder size) effectively tailoring the relative amplitude and phase of the interacting multipoles. Furthermore, these results provide a useful strategy for precise sorting of Ag cylinders by LOFs (Supplementary Fig. 7). In Fig. 3c, we also checked different LOFs induced by materials over a large parameter space of

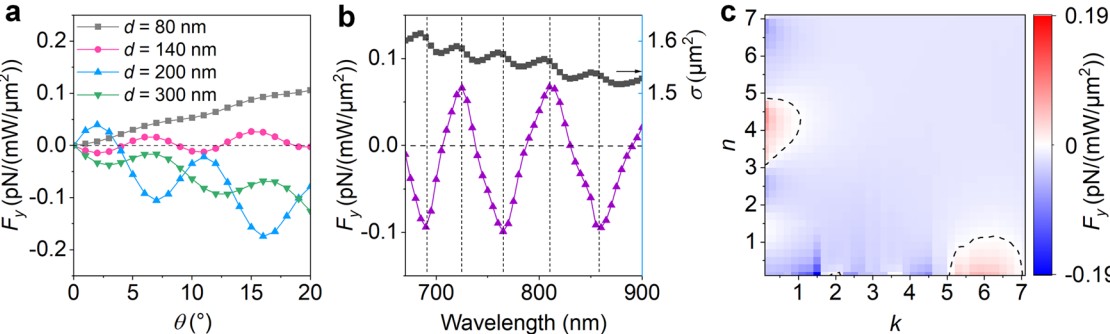

**Fig. 3 | Tuning LOF by the cylinder diameter, the wavelength of light, and material. a** Incident angle-dependent LOF induced in a series of long Ag cylinders with different diameters. **b** Illumination wavelength-dependent LOF applied on a thick Ag cylinder with diameter of 200 nm. As a reference, the black line shows the calculated scattering cross section ($\sigma$) of the same Ag cylinder with different wavelengths. **c** Material-dependent LOF induced in a cylinder with diameter of 200 nm and complex refractive index $n_c = n + ik$. The contours denote $F_y = 0$. The length of the cylinders is fixed at 3900 nm for all calculations. In (**b**, **c**), the incident angle is fixed at 2°.

complex refractive indices ($n_c = n + ik$). We find two parametric regions with reversed sign (see Supplementary Fig. 8 for the creation of different regions of LOF sign reversal by tuning the cylinder diameter), which could be used for material-selective manipulation. For example, it is possible to sort Ag and Au nanowire of the same size (e.g., $d = 200$ nm, $L = 3900$ nm) by a simple plane wave with fixed wavelength (e.g., $\lambda = 800$ nm), which is difficult to achieve with other optical methods[45–48]. Moreover, we also investigated the LOF induced in semiconductor and metal oxide particles (e.g., silicon and zinc oxide nanorods, Supplementary Fig. 9).

## Observation of switchable LOF by holographic optical manipulation of single Ag nanowires

We have established a theoretical framework and predicted a reconfigurable LOF applied on an elongated nanoparticle. To observe this LOF, we developed a flattop phase-gradient optical line (FPOL) for optical manipulation experiments. The optical tweezers system is built with an upright microscope, where the shaped laser beam is focused into a sample cell by an objective. We used a single Ag nanowire as a proof-of-concept demonstration. As shown in Fig. 4a, the optical line has a uniform intensity profile along the long direction (*x* axis). Moreover, in our design, the phase profile of the FPOL, which is defined by a parabolic function $\varphi(x) = 0.2x^2$, gives rise to linearly increased phase gradients ($\nabla\varphi$) along the *x* axis (see Supplementary Fig. 10 for the measured phase profiles). As a result, the FPOL illumination can be considered as a combination of inclined plane wave components (also see Supplementary Fig. 11).

By increasing the phase gradient, the wavevectors of these plane waves are incident at larger inclination angles with the optical axis (*z* axis). When the *x*-position increases by 8 μm, the phase gradient is increased by 3.2 rad/μm. The increased optical phase gradient leads to the creation of a position-dependent LOF which is controlled by polarization angle $\phi$. At the same time, the phase gradient can be used for optically pushing the nanowire to move along the line's orientation[46,49]. As shown in Fig. 4b, when the incident polarization is $\phi = 90°$ (perpendicular to the orientation of the optical line), the force in the lateral direction is dominated by the intensity gradient, with a symmetrical potential well that keeps the nanowire inside it. However, the optical potential wells become asymmetric and even vanish as the LOFs emerge (with diagonal polarizations $\phi = \pm 30°$). Therefore, the nanowire can move along the FPOL until it is released by the increment of the LOF as it progresses along the *x* direction. These results not only show that the FPOL is a reasonable equivalent to obliquely incident plane waves, but also assess FPOL as a flexible and reconfigurable platform for controlled optical transportation and releasing (Fig. 4c), which is not

feasible by using conventional plane waves with a fixed incident angle.

The nanowire has an equilibrium position at $z = 0$ (focal plane) when the radiation pressure (along the *z* axis) is balanced by the electrostatic repulsion. This is done by carefully adjusting the laser power and the surface charges between the nanowire and substrate. Figure 4d shows the dark-field optical images of a single Ag nanowire, which is manipulated by the FPOL with different polarization angles (e.g., ± 30° and 90°). In our experiments, we observed that when the polarization direction changes to ±30°, the orientation of the nanowire either remains parallel to the optical line or can be tuned depending on the magnitude of the phase gradient. When the polarization angle is −30°, the nanowire stably moves several micrometers along the optical line, then it rotates and is released to a specific direction (e.g., +*y*) due to the dominant lateral optical force (e.g., positive LOF) within a carefully designed phase-gradient optical line trap (Supplementary Movie 1). The release direction is reversed by rotating the polarization angle to 30° which reverses the direction of the LOF (Fig. 4e, also see Supplementary Movie 2). Because the LOF vanishes at 90° polarization angle, the nanowire continuously moves along the line until it reaches the end of the FPOL (Supplementary Movie 3, also see Supplementary Fig. 12 for the measured histogram of the orientation of the nanowire in the FPOL). We measured the scattering pattern from a single Ag nanowire under different polarization angles (Supplementary Fig. 13), and the measurements show an asymmetry in scattering which supports the recoil-based origin of the LOF. It is worth noting that in a plane wave with linear polarization parallel to neither the short nor long axis of a cylinder, the cylinder will also rotate due to optical torque (see Supplementary Fig. 14 and Supplementary Note 1)[50,51]. However, in the optical line, the rotation is restricted by the intensity gradient force along the short axis of the trap, and controlled lateral transportation of the cylinder without rotation maybe achieved by using arrays of optical lines[51].

To rule out alternative explanation to the lateral force such as thermophoretic motion in the fluid, we also calculated the temperature field distribution on the nanowire and the surrounding fluid medium (Supplementary Fig. 15). Our simulations show that when using a laser intensity of 4 mW/μm², the maximum temperature increase is ~21 K. Note that the thermophoretic force, which is symmetric with respect to the long axis of the nanowire, does not contribute to the lateral force. Moreover, the sign of the thermophoretic force depends on the sign of the thermo-diffusion coefficient, which is usually positive for metal particles[52]. As a result, changing the polarization does not affect the sign of the thermophoretic force. Thus, the observed effect cannot be due to thermophoretic forces, and the optical force is responsible.

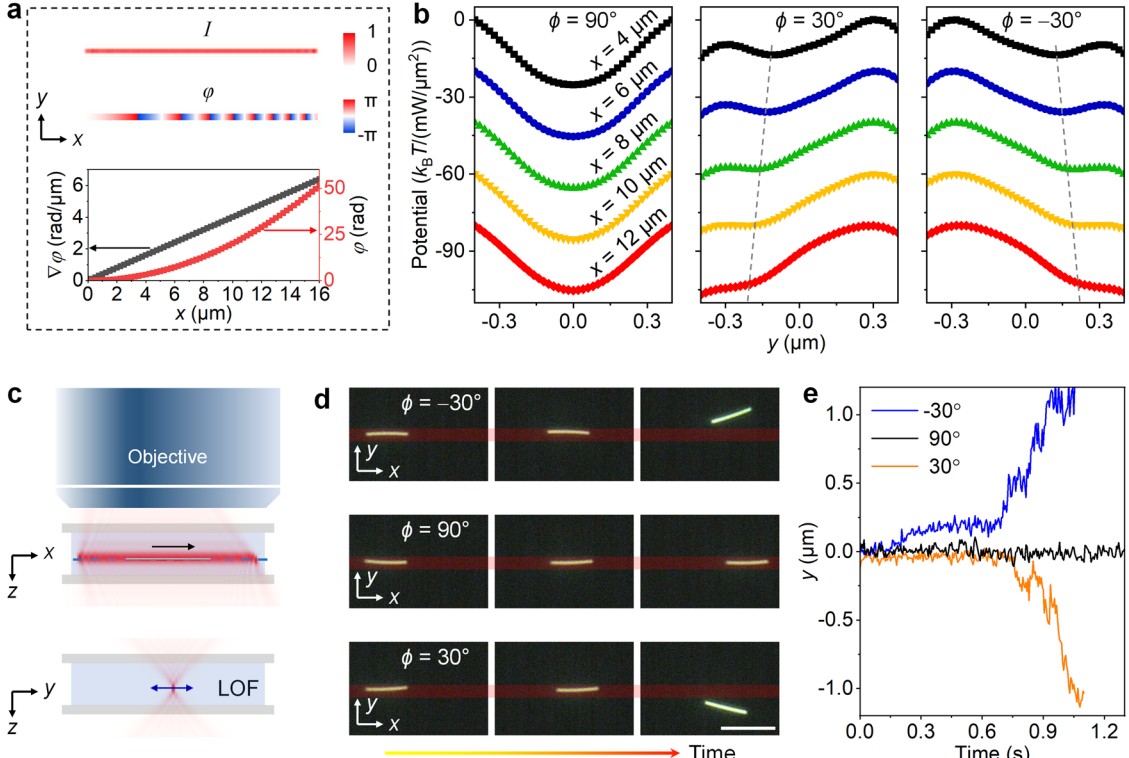

**Fig. 4 | Direct experimental evidence of interface-free LOF. a** Design of a holographic optical field for linear polarization angle-controlled transportation and releasing of single Ag nanowires. The intensity and phase gradients are created along the short ($y$) and long ($x$) axis, respectively. Because the intensity gradients are symmetric in the $y$–$z$ plane, they do not produce additional LOF in our experiments. **b** Calculated potential energy (along the short axis of the optical line) of a single Ag cylinder ($L = 3900$ nm, $d = 80$ nm) at different $x$-positions of the optical field. The $x$-positions (marked by different colors) are 4, 6, 8, 10, 12 μm, respectively. The depth and symmetry of the potential well are dependent of the linear polarization angle and optical phase gradient. **c** Illustration of the $x$–$z$ and $y$–$z$ views of the optical line trap. The nanowire can move along the optical line due to the phase-gradient forces. The short-axis intensity gradient keeps the orientation of the nanowire which may escape with increased LOF along its moving direction. **d, e** Demonstration of switchable LOF with different linear polarization angles. The polarization angle is defined as the angle with respect to $x$ axis. The red strips, overlaid on the (**d**) dark-field images of a single Ag nanowire, represent the optical line. The scale bar is 5 μm and the laser wavelength is 800 nm. **e** Measured $y$-trajectories of the Ag nanowire under three different polarization angles.

## Discussion

In summary, we have revealed that the multipolar interplay in a nonchiral elongated nanoparticle can produce an interface-free LOF even under the illumination of a linearly polarized plane wave. The applicability of different LOF systems is summarized in Table 1. We find that the sign and magnitude of this multipolar interplay-based LOF can be flexibly tailored by the polarization direction, incidence angle and wavelength of the optical field and the aspect ratio and material of the particle. These findings, in turn, provide a promising strategy for optomechanical applications such as single-particle optical sorting and light-driven micro/nanomotors. Furthermore, direct experimental evidence of switchable LOF is presented in a holographic optical manipulation platform, which provides a practical strategy for optical trapping and manipulation of elongated nanoparticles.

For a single Ag cylinder, the spatial distribution of the energy of the surface plasmon can be tailored by varying the polarization angle. A chiral surface plasmon is excited due to the coherent superposition of transverse and longitudinal plasmon modes[53]. We note that the sign of the LOF and the handedness of the chiral surface plasmons are correlated because they can reverse simultaneously when switching the polarization angle between positive and negative (Supplementary Fig. 16). This is an analogy to a previous observation that chiral particle-substrate interactions can lead to LOF, where the sign of optical force is switchable by tuning the handedness of the particle[6]. Moreover, the magnitude of the induced LOF in linear polarization is comparable to that of the LOF in circular polarization (Supplementary Fig. 17). However, it should be noted that the LOF reported here is produced by a free-standing nonchiral nanostructure (i.e., in the absence of particle–substrate interaction). The addition of a surface could potentially increase the magnitude of this LOF, due to the near-field directional excitation of surface modes[24,54].

It is worth noting that a single metal cylinder also experiences an optical pulling force[33,34,55,56] at the same time (also see Supplementary Fig. 18), which is comparable in magnitude to the LOF. The forward scattering (along $x$ axis) can create a strong recoil force. For example, with a length of 575 nm (Fig. 2d), this recoil force is much stronger than the extinction force, thus optical pulling is achieved. We envision that the elongated particle can act as a plane wave-driven optical micro-nanomotor[26–28]. It would be able to move along complex paths in a two-dimensional plane with two independent control channels. These control channels are established by tuning the polarization and incident angle, where forward and backward movements are reversed by optical pushing and pulling, while left–right translations are controlled by switchable LOF. Moreover, the radiation pressure (along the $z$ axis) can be compensated by using two counter-propagated plane waves with linear polarization. Thus, it is possible to extend our manipulation paradigm from two dimensions (near a substrate) to three dimensions (also see Supplementary Fig. 19).

While this work mainly focuses on single nanowires, the principle of creating lateral optical forces through multipolar interplay can be extended to the design of micro/nanomotors with nanowire arrays on substrates[57]. Various nanofabrication techniques have been developed to pattern metal nanowires on a substrate, either horizontally by photolithography[58] or E-beam lithography[59], or vertically by

**Table 1 | Applicability of different LOF systems**

| System | Substrate-free platform | Plane wave-like illumination | Nonchiral material | Homogeneous material |
|---|---|---|---|---|
| Material chirality–spin coupling[6] | No | Yes | No | Yes |
| Transverse spin[9] | No | No | No | Yes |
| Spin–orbit coupling[8] | No | Yes | Yes | Yes |
| Transverse momentum[7,11,16,21,25] | Yes | No | Yes | Yes |
| Metamaterial[26–28] | Yes | Yes | Yes | No |
| Multipolar interplay (this work) | Yes | Yes | Yes | Yes |

membrane template-assisted electrodeposition[57]. The substrate can provide a mechanical support to the nanowires, where the lateral optical forces on individual nanowires can be accumulated, leading to strong driving forces to achieve optical metavehicles[26–28].

## Methods

### Numerical simulations

Radiation patterns from electric and magnetic dipoles are simulated by using the software Lumerical FDTD Solutions. To calculate the LOF induced in the Ag cylinder, a TFSF (total-field scattered field, $\lambda = 800$ nm) source (with tunable incident and polarization angles) was used. The cylinder was immersed in water and placed in the $x$–$y$ plane with its $z$-position fixed. The optical force is obtained by integrating the time-averaged Maxwell stress tensor over a surface surrounding the cylinder. The linearly polarized optical line ($\lambda = 800$ nm) with arbitrary intensity and phase profiles can be introduced by using the "Import Source" function. The optical potential is obtained by integrating the optical force ($F_y$) over the $y$ axis.

### Multipolar interplay analysis

In Mie theory, the scattered fields are expanded in terms of vector spherical wave functions[39,60]

$$\mathbf{E}_{\text{sca}}(\mathbf{x}) = E_0 \sum_{l=1}^{\infty} \sum_{m=-l}^{l} [a_{\text{M}}(l,m)\mathbf{M}_{lm}(k\mathbf{x}) + a_{\text{E}}(l,m)\mathbf{N}_{lm}(k\mathbf{x})], \quad (10)$$

where

$$\left.\begin{array}{r}\mathbf{M}_{lm}(k\mathbf{x})\\\mathbf{N}_{lm}(k\mathbf{x})\end{array}\right\} = D_{lm}\left\{\begin{array}{c}h_l(kr)\mathbf{C}_{lm}(\theta,\phi)\\\frac{l(l+1)}{kr}h_l(kr)\mathbf{P}_{lm}(\theta,\phi) + \frac{[krh_l(kr)]'}{kr}\mathbf{B}_{lm}(\theta,\phi)\end{array}\right., \quad (11)$$

with

$$D_{lm} = \sqrt{\frac{(2l+1)(l-m)!}{4\pi(l+1)(l+m)!}} \quad (12)$$

In the above formulae, the vector spherical harmonic functions (VSHFs) are defined as

$$\mathbf{P}_{lm}(\theta,\phi) = \mathbf{e}_r P_l^m(\cos\theta)e^{im\phi},$$

$$\mathbf{B}_{lm}(\theta,\phi) = \left[\mathbf{e}_\theta \frac{d}{d\theta}P_l^m(\cos\theta) + \mathbf{e}_\phi \frac{im}{\sin\theta}P_l^m(\cos\theta)\right]e^{im\phi}, \quad (13)$$

$$\mathbf{C}_{lm}(\theta,\phi) = \left[\mathbf{e}_\theta \frac{im}{\sin\theta}P_l^m(\cos\theta) - \mathbf{e}_\phi \frac{d}{d\theta}P_l^m(\cos\theta)\right]e^{im\phi}.$$

The relation between $\mathbf{C}_{lm}$ and the VSHF $\mathbf{X}_{lm}$ is given by $D_{lm}\mathbf{C}_{lm} = i^l[\pi(2l+1)]^{1/2}\mathbf{X}_{lm}$.

One may extend the multipolar expansion to the field in the far-field region ($kr \to \infty$) so that it is applicable to our far-field scattering patterns. This is achieved by using the asymptotic expression for the spherical Hankel function of the first kind:

$$h_l(kr) \underset{kr\to\infty}{=} (-i)^{l+1}\frac{e^{ikr}}{kr}. \quad (14)$$

By doing so, we are able to express the far-field scattered field as

$$\mathbf{E}_{\text{sca}}(\mathbf{x}) = \frac{E_0 e^{ikr}}{kr}\sum_{l=1}^{\infty}\sum_{m=-l}^{l}(-i)^l D_{lm}[-ia_{\text{M}}(l,m)\mathbf{C}_{lm}(\theta,\phi)$$
$$+ a_{\text{E}}(l,m)\mathbf{B}_{lm}(\theta,\phi)] \equiv \frac{e^{ikr}}{kr}\mathcal{E}_{\text{sca}}(\theta,\phi), \quad (15)$$

where $\mathcal{E}_{\text{sca}}(\theta,\phi)$ denotes the angle-dependent part of the far-field electric vector. Eventually, with the orthogonality relations:

$$\int_\Omega\left[\mathbf{B}_{lm}(\theta,\phi)\cdot\mathbf{C}_{lm}^*(\theta,\phi)\right]d\Omega = 0, \int_\Omega\left[\mathbf{B}_{lm}(\theta,\phi)\cdot\mathbf{B}_{l'm'}^*(\theta,\phi)\right]d\Omega$$
$$= \int_\Omega\left[\mathbf{C}_{lm}(\theta,\phi)\cdot\mathbf{C}_{l'm'}^*(\theta,\phi)\right]d\Omega = \frac{1}{(D_{lm})^2}\delta_{ll'}\delta_{mm'}, \quad (16)$$

we have

$$a_{\text{M}}(l,m) = \frac{iD_{lm}}{(-i)^l}\int_\Omega[\mathbf{C}_{lm}^*(\theta,\phi)\cdot\mathcal{E}_{sca}(\theta,\phi)]d\Omega,$$
$$a_{\text{E}}(l,m) = \frac{iD_{lm}}{(-i)^l}\int_\Omega[\mathbf{B}_{lm}^*(\theta,\phi)\cdot\mathcal{E}_{sca}(\theta,\phi)]d\Omega, \quad (17)$$

for the magnetic and electric scattering coefficients. Equation (17) offers a universal method to evaluate the scattering coefficients with the field vector in the far zone.

To test the validity of the method, we calculate the scattering coefficients for the nanorod with $L = 350$ nm, where $\mathcal{E}_{\text{sca}}(\theta,\phi)$ is obtained from the FDTD simulation (Supplementary Fig. 20a). The coefficients for $l \geq 3$ are negligible, thus the highest mode supported by this particle is $l = 2$ (i.e., quadrupole). The total radiation intensity distribution (see Supplementary Fig. 20b) is calculated via these coefficients and Eq. (15), which agrees well with our FDTD simulation results (Fig. 2d, second row). Note that in the calculation of the total intensity $|\mathcal{E}_{\text{sca}}(\theta,\phi)|^2$, there are product terms of the coefficients with different orders or types, which represent the contributions from multipolar interference. For example, the summation of the terms related to $a_{\text{E}}(1,m)a_{\text{E}}^*(2,m')$ denotes the ED-EQ contribution. These interference components can be either positive or negative; they do not contribute to the scattering cross section due to the orthogonal relations (Eq. (16)), but may result in the asymmetric scattering and hence, the recoil optical force.

### Instrumentation and materials

The Ag nanowires were purchased from NanoSeedz. The optical tweezer system is built within a home-built microscope. A collimated laser beam (800 nm, Spectra-Physics 3900S), shaped by a spatial light modulator (LCOS-SLM, Hamamatsu, X13138), is focused into a sample cell by a super apochromat objective (Olympus, UPLSAPO60XW). The flattop phase-gradient optical line trap is created by a computer-generated holography (CGH) method we previously developed[30], which is based on superimposing an array of diffraction-limited focal spots with designed phase. The shaped incident optical field can be

written as:

$$U(x,y) = \sum_{n=1}^{N} A \exp\left[\frac{2\pi}{\lambda f}(xy_n + yx_n)i + \phi_n i\right], \qquad (18)$$

where $N$ is the number of the focal spots, $(x_n, y_n)$ are the coordinates relative to the original focal center (0, 0), $f$ is the focal length of the objective, $\lambda$ is the laser wavelength, $\phi_n$ is the attached phase, and $A$ modifies the intensity profile of the optical line trap. Then it is transformed into an analytical formula to obtain a digital hologram. Motion of the Ag nanowire is recorded by a CMOS camera (Point Grey Grasshopper3). A beam profiler and a wavefront sensor are used to measure the intensity and phase profile of the optical trap, respectively.

## Data availability
The data of the main text are provided in the Source Data file. Additional data related to this paper may be requested from the authors. Source data are provided with this paper.

## Code availability
The code used in this work is available from the corresponding authors upon request.

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

## Acknowledgements

This paper is dedicated to the memory of our dear co-worker Zijie Yan, who passed away while this paper was being peer-reviewed. We thank Dr. Huanjun Chen and Tianyue Li for useful discussions. F.N. acknowledged the support from the startup funding in Jinan University. This work was supported by the Guangdong Basic and Applied Basic Research Foundation (Nos. 2023A1515011330 and 2023A1515030143), National Natural Science Foundation of China (Nos. 12274181, 11974417, 62135005, 11804119, and 12074169), Guangdong Province Talent Recruitment Program (No. 2021QN02C103), Stable Support Plan Program of Shenzhen Natural Science Fund (No. 20200925152152003), and Guangzhou Science and Technology Program (No. 2023A04J1059). J.J.K.-S. and F.J.R.-F. acknowledge participation in EIC-Pathfinder-CHIRALFORCE (101046961) funded by Innovate UK Horizon Europe Guarantee (UKRI project 10045438).

## Author contributions

F.N. and X.X. conceived the research. F.N. discovered the lateral force system, performed the numerical calculations, and designed the optical manipulation experiments. X.X. developed the analytical model to interpret the physics of lateral force. S.Y. and X.X. did the multipolar interplay analysis. F.J.R.-F., J.J.K.-S., J.N., and B.Y. were involved in the discussion and analysis. F.N., X.X., S.Y., and Z.Y. oversaw and directed the whole project. All authors contributed to the preparation of the manuscript.

## Competing interests

The authors declare no competing interests.
