## [Peer Review File · Nature Communications]

REVIEWER COMMENTS

Reviewer #1 (Remarks to the Author):

Traditional optical trapping employs the intensity gradient of a strongly focused light beam. In recent years, attempts were made to achieve optical trapping/manipulation without intensity gradient, and the lateral optical force (LOF) considered in this manuscript is a promising candidate. The effectiveness of LOT often relies heavily on the use of tailored objects and beams.

In this manuscript, the authors demonstrated LOFs on elongated nanoparticles tunable (both sign and magnitude) by a linearly polarized plane wave. An analytical model that elucidates the underlying physics is also provided. The experimental investigations conducted by the authors employ a flat-top phase-gradient optical line trap. Through this setup, they successfully demonstrate controlled transportation and bidirectional releasing of a single Ag nanowire using polarization angle-controlled LOFs.

In my opinion, the manuscript is innovative and may open up a novel way to optical manipulation. I would recommend the manuscript to be published in Nature Communications, given that the authors can address my concerns listed below.

1. The radiation diagrams in Figure 1 would benefit from labeling the axes.
2. The prefactor of the magnetoelectric interference force F_{em} [Eq.(1)] is inconsistent with that in Ref.[30]. This discrepancy should be clarified for the sake of readers. Is it caused by the use of different units?
3. As the paper deals with monochromatic fields, the assumption of the time dependence $\exp(-i\omega t)$ in Eq. (4) should be mentioned.
4. It might be valuable for the authors to provide some insights or comments on the possibility of nanorods/nanowires rotating with linear polarization.
5. The experimental investigations conducted by the authors employ a clever setup involving a flat-top phase-gradient optical line trap. The authors should provide more details in designing of this kind of optical trap.
6. The nanomanipulation experiments were conducted near a substrate. Can the approach be extended to 3D manipulation, for example, in a liquid? It would be valuable to discuss the feasibility and challenges associated with such extensions.
7. Considering the principles and techniques outlined in this study, it raises the question of whether they can be applied to the design and fabrication of micro/nanomotors using alternative methods, such as E-beam lithography. Exploring the potential application of the reported principles in different fabrication techniques would enhance the practicality and versatility of this research. The authors could discuss the

compatibility and potential advantages or limitations of implementing their approach in conjunction with E-beam lithography or other relevant techniques for micro/nanomotor fabrication. This discussion would further enrich the scope and impact of the study.

Reviewer #2 (Remarks to the Author):

The authors report a lateral optical force acting on elongated nanoparticles illuminated by an obliquely incident plane wave. Since the sign and magnitude of the lateral force depend on the particle parameters such as the cylinder diameter and complex refractive index, this force has potential applications in optical sorting. Furthermore, it is interesting to note that the sign of the transverse force switches with the polarization of the incident light, which is also supported by the designed experiment for an Ag nanowire.

However, there are concerns about the physics underlying the lateral optical force. The author argued that the lateral force result from asymmetric side scattering due to the magnetoelectric interplay of longitudinal and transverse modes, which is one of key points in this manuscript. However, the evidence of this mechanism is not sufficient. In the previous studies, e.g., D. Vercruysse et al, *Nano Lett*, 13, 3843-3849 (2013), the interference between electric quadrupole and dipole modes in the metal nanoparticles also results in the asymmetric side scattering. Indeed, the obliquely incident plane wave in Figure 1c allows to excite the electric quadrupole mode of the elongated metal nanoparticles. Therefore, in order to strongly support the mechanism based on the magnetoelectric interplay, mode expansion analysis of the scattered field is essential, as shown in the previous papers, e.g., J. Li et al, *Nano Lett*, 16, 4396-4403 (2016).

Furthermore, the authors should address the following comments to improve the manuscript.

- (1) To support the switchable lateral optical force in Figure 4, the authors should measure the scattering pattern from the nanowire at different linear polarization angle of an incident light.
- (2) What is the temperature rise on the Ag nanowires in Figure 4? Given the strong photothermal response of Ag nanowires, it is possible that an appreciable temperature rise could occur, which may give rise to thermophoretic motion in the fluid. The authors should compute the temperature field distribution on the nanowire and the surrounding fluid medium for the laser intensity used in the experiment. Is there any thermophoretic force on the nanowire? The sign of the thermophoretic force does not change depending on the incident polarization angle?
- (3) Why do not you submit the real-time movies that shows the lateral force switched by the incident polarization angle in Figure 4d? They would be very insightful for the readers.

Reviewer #3 (Remarks to the Author):

Referee report on manuscript NCOMMS-23-12101, "Creating tunable optical forces..."

This paper deals with the lateral optical forces on metallic nanowires (Ag) of different aspect ratios. The work is both theoretical and experimental.

The authors show through calculations that a silver nanowire illuminated by a plane wave, hence an optical field that is homogeneous in space, still can produce a force sideways, in a direction perpendicular to the propagation direction of the light. For this to occur some conditions must be met that produce a lowering of the symmetry. The wave must hit the nanowire at an angle of incidence that differs from zero and moreover, the incident light, while linearly polarized, must consist of a mixture of p and s polarization. The sideways, lateral, force is the result of the recoil forces that occur when photons are predominantly scattered in one of the lateral directions due to an interference between the magnetic and electric multipoles of the nanoparticle. The authors also show experimental results that support the theoretical results.

I find the basic results and conclusions presented in the manuscript to be highly believable and scientifically sound. I also think that the discussion and results presented in the manuscript will inspire further research in this field. The paper contains a solid list of very relevant references to the literature concerning the optical forces.

I think the authors have used methods that are appropriate in conducting the research. The theoretical work is built on a combination of a simple dipole model that clearly illustrates what is happening, and detailed numerical calculations using the FDTD method. I myself am not sufficiently knowledgeable about the experimental methods to assess the experimental methods used in detail.

I can identify one apparent weakness in terms of the applicability of this work. It has to do with the fact that the authors consider a very specific configuration of the nanowire in which it is always oriented with its long axis in the same plane as the wave vector of the incident light (the plane of incidence). It is not unlikely that fluctuations and/or optical torques would turn the nanowire away from that position and the manuscript does not address this question at any length. In Figure 4 concerning the optical sorting experiment that the authors have carried out one can see that the nanowires to begin with stay oriented along the optical line but that they eventually leave that line (which happens by design), and then turn away from the original orientation. Hence, I think the manuscript would be strengthened by a discussion concerning the robustness of the orientational alignment of the nanowire.

I find the paper to be easy to read for the most part, but I have some suggestions for improvements, primarily concerning the figures.

The results for the lateral force as a function of incidence angle in Fig. 1d (the right panel) shows some irregular oscillations. What is the reason for this? In the case of the long wire I can see that there are computational challenges involved, but I do not see why that would also be the case for the short wire.

In figure 1 a and 1 b, I lack a clear explanation of what the red and blue color mean in the radiation lobes there. I think the caption needs to be more specific there.

The same also goes for Fig. 2c and 2d. Again I do not fully understand what the colors mean. Figure 2c also contains a mistake in that the angle of incidence there is 2 degrees (as the caption says) rather than 0 degrees (as the actual figure says). Finally, concerning Fig 2e the authors talk about a logarithmic scale but it is unclear whether that has been used in producing Fig. 2e.

On line 175 and 176 the authors state that the radiation pattern for a short wire (100 nm) shows an "abnormal" dumbbell shape. I would say that that shape is quite normal for dipole radiation.

To summarize, I think the paper is scientifically sound and very interesting but in need of a more detailed discussion and explanations on some points.

Response

We thank the Editor and Reviewers for the valuable time. The reviewers' thoughtful comments are answered point-by-point below, which have significantly strengthened our manuscript (NCOMMS-23-12101). For convenience, we uploaded a second copy of the manuscript as Supplementary Material, where all revisions are highlighted in red.

Reviewer 1

Traditional optical trapping employs the intensity gradient of a strongly focused light beam. In recent years, attempts were made to achieve optical trapping/manipulation without intensity gradient, and the lateral optical force (LOF) considered in this manuscript is a promising candidate. The effectiveness of LOT often relies heavily on the use of tailored objects and beams.

In this manuscript, the authors demonstrated LOFs on elongated nanoparticles tunable (both sign and magnitude) by a linearly polarized plane wave. An analytical model that elucidates the underlying physics is also provided. The experimental investigations conducted by the authors employ a flat-top phase-gradient optical line trap. Through this setup, they successfully demonstrate controlled transportation and bidirectional releasing of a single Ag nanowire using polarization angle-controlled LOFs.

In my opinion, the manuscript is innovative and may open up a novel way to optical manipulation. I would recommend the manuscript to be published in Nature Communications, given that the authors can address my concerns listed below.

We thank the reviewer for his/her recommendation and for recognizing that our work may open a novel way to optical manipulation.

Comment 1: *The radiation diagrams in Figure 1 would benefit from labeling the axes.*

Reply 1: We have added x and y axes in the revised Figure 1.

Comment 2: *The prefactor of the magnetoelectric interference force F_{em} [Eq.(1)] is inconsistent with that in Ref.[30]. This discrepancy should be clarified for the sake of readers. Is it caused by the use of different units?*

Reply 2: Yes, throughout the paper we use SI units while Gaussian units are employed in Ref. [30]. We clarify this in the revised manuscript.

Comment 3: *As the paper deals with monochromatic fields, the assumption of the time dependence $\exp(-i\omega t)$ in Eq. (4) should be mentioned.*

Reply 3: We now mention this in the revised manuscript.

Added in main text (Page 6): "...where we have assumed the time dependence $\exp(-i\omega t)$."

Comment 4: *It might be valuable for the authors to provide some insights or comments on the possibility of nanorods/nanowires rotating with linear polarization.*

Reply 4: We have discussed the possibility of nanorods/nanowires rotating with linear polarization in the revised manuscript.

Added in main text (Page 13): It is worth noting that in a plane wave with linear polarization parallel to neither the short nor long axis of a cylinder, the cylinder will also rotate due to optical torque (see Fig. S14 and Supplementary Note 1)^{48,49}. However, in the optical line, the rotation is restricted by the intensity gradient force along the short axis of the trap, and controlled lateral transportation of the cylinder without rotation may be achieved by using arrays of optical lines⁴⁹.

Added a new section in the supplementary information: Note 1. Optical rotation induced by linear polarization

To figure out if a nanorod could rotate, we checked the optical torque applied on a series of Ag nanocylinders with different lengths (their diameters are fixed at 80 nm). In the numerical calculations shown in the main text, the total optical force (F_y) is obtained by integrating the Maxwell stress tensor over a surface surrounding the whole cylinder. This surface can be divided into two equal parts, then the optical torque for the entire cylinder is the sum of the torque on each part relative to the cylinder center. When the polarization direction is parallel or perpendicular to the orientation of the cylinders, they do not experience any optical torque. When the polarization angle is 45° , a cylinder with length of 100 nm shows a significant preference to align parallel to the polarization direction. As the cylinder length increases to 200 nm, it prefers to align perpendicular to the polarization direction. Fig. S14 shows the lateral force and torque applied on a long nanowire ($L = 3900$ nm) at different incident angles.

Supplementary Figure 14. Optical rotation induced by linear polarization. Calculated optical torque and lateral force of a long cylinder illuminated with different incident angles. The polarization angle is 45° . The cylinder's rotation direction may vary at different incident angles.

Comment 5: *The experimental investigations conducted by the authors employ a clever setup involving a flattop phase-gradient optical line trap. The authors should provide more details in designing of this kind of optical trap.*

Reply 5: We have provided more details in designing the flattop phase-gradient optical line trap in the revised manuscript.

Added in Method (Page 17): The flattop phase-gradient optical line trap is based on a computer-generated holography (CGH) method we previously developed²⁹, which is calculated by superimposing an array of diffraction-limited focal spots with designed phase. The shaped incident optical field can be written as:

$$U(x, y) = \sum_{n=1}^N A \exp\left[\frac{2\pi}{\lambda f} (xy_n + yx_n)i + \phi_n i\right], \quad (14)$$

where N is the numbers of the focal spots, (x_n, y_n) are the coordinates relative to the original focal center $(0, 0)$, f is the focal length of the objective, λ is the laser wavelength, ϕ_n is the attached phase, and A modifies the intensity profile of the optical line trap. Then it is transformed into an analytical formula to obtain a digital hologram.

Comment 6: *The nanomanipulation experiments were conducted near a substrate. Can the approach be extended to 3D manipulation, for example, in a liquid? It would be valuable to discuss the feasibility and challenges associated with such extensions.*

Reply 6: Our optical manipulation paradigm can be extended to three dimensions. Reversible and tunable LOF in a liquid can be achieved by using two counter-propagated plane waves with linear polarization.

Added in main text (Page 15): Moreover, the radiation pressure (along the z -axis) can be compensated by using two counter-propagated plane waves with linear polarization. Thus, it is possible to extend our manipulation paradigm from two dimensions (near a substrate) to three dimensions (also see Fig. S19).

Supplementary Figure 19. Reversible and tunable LOF excited by two counter-propagated plane waves with linear polarization. (a) Calculated LOF (F_y) as a function of incident angle of two plane waves. The LOF takes opposite sign by switching the sign of the polarization angle ϕ . (b) Calculated LOF as a function of the polarization angle of an oblique plane wave ($\lambda = 800$ nm, $\theta = 15^\circ$). (c,d) The corresponding optical force (F_z) along z direction. The diameter and length of the Ag cylinder are 80 and 1500 nm, respectively. The F_y and F_z created by a single plane wave with the same incident and polarization angles are also shown for comparison (plotted by dashed lines).

Comment 7: *Considering the principles and techniques outlined in this study, it raises the question of whether they can be applied to the design and fabrication of micro/nanomotors using alternative methods, such as E-beam lithography. Exploring the potential application of the reported principles in different fabrication techniques would enhance the practicality and versatility of this research. The authors could discuss the compatibility and potential advantages or limitations of implementing their approach in conjunction with E-beam lithography or other relevant techniques for micro/nanomotor fabrication. This discussion would further enrich the scope and impact of the study.*

Reply 7: We thank the reviewer for the suggestions. We have added a paragraph to discuss a potential way for the design and fabrication of nanorod-based micro/nanomotors by taking advantage of LOF.

Added in main text (Page 15): While this work mainly focuses on single nanowires, the principle of creating lateral optical forces through multipolar interplay can be extended to the design of micro/nanomotors with nanowire arrays on substrates⁵⁶. Various nanofabrication techniques have been developed to pattern metal nanowires on a substrate, either horizontally by photolithography⁵⁷ or E-beam lithography⁵⁸, or vertically by membrane template-assisted electrodeposition⁵⁶. The substrate can provide a mechanical support to the nanowires, where the lateral optical forces on individual nanowires can be accumulated, leading to strong driving forces to achieve new types of optical metavehicles²⁵⁻²⁷.

Reviewer 2

The authors report a lateral optical force acting on elongated nanoparticles illuminated by an obliquely incident plane wave. Since the sign and magnitude of the lateral force depend on the particle parameters such as the cylinder diameter and complex refractive index, this force has potential applications in optical sorting. Furthermore, it is interesting to note that the sign of the transverse force switches with the polarization of the incident light, which is also supported by the designed experiment for an Ag nanowire.

We thank the reviewer for his/her positive evaluation.

Comment 1: *However, there are concerns about the physics underlying the lateral optical force. The author argued that the lateral force result from asymmetric side scattering due to the*

magnetolectric interplay of longitudinal and transverse modes, which is one of key points in this manuscript. However, the evidence of this mechanism is not sufficient. In the previous studies, e.g., D. Vercruyse et al, *Nano Lett*, 13, 3843-3849 (2013), the interference between electric quadrupole and dipole modes in the metal nanoparticles also results in the asymmetric side scattering. Indeed, the obliquely incident plane wave in Figure 1c allows to excite the electric quadrupole mode of the elongated metal nanoparticles. Therefore, in order to strongly support the mechanism based on the magnetolectric interplay, mode expansion analysis of the scattered field is essential, as shown in the previous papers, e.g., J. Li et al, *Nano Lett*, 16, 4396-4403 (2016).

Reply 1: We thank the reviewer for raising this important issue. We agree with the reviewer that quadruple may exist in single nanocylinders. In fact, as we can see from Fig. 2d, the radiation pattern for a cylinder with a length of 350 nm exhibits four petals, which is the feature of electric quadrupole radiation. The number of petals increases for longer nanorods, revealing the existence of higher multipole modes. As per the reviewer’s suggestion, we have performed a mode expansion analysis with multipole theory reported by Grahn et al. (which is also used in the two references mentioned by the reviewer) and added new calculation results in Fig. 2e. These new results clearly indicate that the physics underneath of the LOF is not limited to the dipolar landscape. Higher multipoles are also exploitable for creating this force. In the revised manuscript, we have cited these references and highlighted the importance of the multipolar interference in the asymmetric scattering and the LOF. We also modify the title to “*Creating tunable lateral optical forces through multipolar interplay in single nanowires*”, for the sake of comprehensiveness, accuracy, and potential interest.

Fig. 2. Tuning LOF by cylinder length. a Calculated incident angle-resolved scattering cross section of a Ag cylinder with different length L . The abrupt changes shown in the scattering

spectrum are caused by the optical resonance. The inset shows the scattering cross sections at two representative angles. **b** Calculated LOFs on the Ag cylinder with different lengths. The inset shows the LOFs at two representative angles. **c, d** Normalized far-field angular distributions of the scattered intensity of selected Ag cylinders with (**c**) small (2°) and (**d**) large angles (20°) of incidence. **e** The absolute values of the radiation intensity distribution resulting from the interaction between different multipoles calculated for $L = 350$ nm. Dashed and solid lines represent the negative and positive contributions, respectively, due to the destructive and constructive interference. For all calculations, the cylinder diameter and the polarization angle are fixed at 80 nm and 45° , respectively.

Added in main text (Page 8): To figure out if the LOF is restricted to the dipolar landscape, we perform a mode expansion analysis with multipole theory³⁷⁻⁴¹. Because the radiation of single multipoles is always symmetric³⁹, the asymmetric scattering must originate from the interaction between different multipoles. Therefore, we focus on the radiation associated with the interference terms of the scattering coefficients with different order l or type (electric/magnetic) (also see Methods). As a representative demonstration, Fig. 2e shows the radiation patterns created by different interacting modes in the Ag cylinder with $L = 350$ nm. In contrast to the total scattering intensity which is always positive (Fig. 2d), these interplay-related components can be negative in some directions (indicated by dashed lines), suppressing the net scattering due to destructive interference, as explained in the Methods section.

For the ED-MD interference (the first row), the scattering intensity is mainly negative (or destructive) in the lower half arc, while being positive (or constructive) in the upper. From such asymmetric scattering, one can expect the emergence of a recoiling force in the lateral direction, which has already been captured by our dipole model (Eq. (5)). Remarkably, the constructive and destructive interference can also occur when the electric quadrupole (EQ) interacts with the magnetic quadrupole (MQ) or ED, resulting in an unbalanced radiation from which the LOF can be expected. The same is true for the MD-MQ interplay, whose scattering intensity, despite small, exhibits non-symmetric angular dependence. However, the interplay between ED-MQ or EQ-MD results in a symmetric profile, which is not responsible for the LOF. This is in agreement with the Lorenz-Mie theory that the recoiling optical force on a single particle can only stem from the interaction between hybrid (magnetolectric) modes with the same order, or purely electric (or magnetic) modes of adjacent orders⁴². Similar LOF mechanism can be expanded beyond the quadrupoles. For example, asymmetric radiation profiles can be produced by the octupoles excited in longer nanorods (e.g., $L = 575$ nm, Fig. S6).

Supplementary Figure 6. Multipolar interplay analysis. A similar plot for the octupole-induced asymmetric radiation in the cylinder with $L = 575$ nm. EO, electric octupole; MO, magnetic octupole.

Added in method section: Multipolar interplay analysis. In Mie theory, the scattered fields are expanded in terms of vector spherical wave functions^{37,59}

$$\mathbf{E}_{\text{sca}}(\mathbf{x}) = E_0 \sum_{l=1}^{\infty} \sum_{m=-l}^l [a_M(l, m) \mathbf{M}_{lm}(k\mathbf{x}) + a_E(l, m) \mathbf{N}_{lm}(k\mathbf{x})], \quad (6)$$

where

$$\left. \begin{array}{l} \mathbf{M}_{lm}(k\mathbf{x}) \\ \mathbf{N}_{lm}(k\mathbf{x}) \end{array} \right\} = D_{lm} \left\{ \begin{array}{l} h_l(kr) \mathbf{C}_{lm}(\theta, \phi) \\ \frac{l(l+1)}{kr} h_l(kr) \mathbf{P}_{lm}(\theta, \phi) + \frac{[kr h_l(kr)]'}{kr} \mathbf{B}_{lm}(\theta, \phi) \end{array} \right\}, \quad (7)$$

with

$$D_{lm} = \sqrt{\frac{(2l+1)(l-m)!}{4\pi l(l+1)(l+m)!}}. \quad (8)$$

In the above formulae, the vector spherical harmonic functions (VSHFs) are defined as

$$\begin{aligned} \mathbf{P}_{lm}(\theta, \phi) &= \mathbf{e}_r P_l^m(\cos \theta) e^{im\phi}, \\ \mathbf{B}_{lm}(\theta, \phi) &= \left[\mathbf{e}_\theta \frac{d}{d\theta} P_l^m(\cos \theta) + \mathbf{e}_\phi \frac{im}{\sin \theta} P_l^m(\cos \theta) \right] e^{im\phi}, \\ \mathbf{C}_{lm}(\theta, \phi) &= \left[\mathbf{e}_\theta \frac{im}{\sin \theta} P_l^m(\cos \theta) - \mathbf{e}_\phi \frac{d}{d\theta} P_l^m(\cos \theta) \right] e^{im\phi}. \end{aligned} \quad (9)$$

The relation between \mathbf{C}_{lm} and the VSHF \mathbf{X}_{lm} is given by $D_{lm} \mathbf{C}_{lm} = i^l [\pi(2l+1)]^{1/2} \mathbf{X}_{lm}$.

One may extend the multipolar expansion to the field in the far-field region ($kr \rightarrow \infty$) so that it is applicable to our far-field scattering patterns. This is achieved by using the asymptotic expression for the spherical Hankel function of the first kind:

$$h_l(kr) \underset{kr \rightarrow \infty}{=} (-i)^{l+1} \frac{e^{ikr}}{kr}. \quad (10)$$

By doing so, we are able to express the far-field scattered field as

$$\begin{aligned} \mathbf{E}_{\text{sca}}(\mathbf{x}) &= \frac{E_0 e^{ikr}}{kr} \sum_{l=1}^{\infty} \sum_{m=-l}^l (-i)^l D_{lm} [-ia_M(l, m) \mathbf{C}_{lm}(\theta, \phi) + a_E(l, m) \mathbf{B}_{lm}(\theta, \phi)] \\ &\equiv \frac{e^{ikr}}{kr} \mathcal{E}_{\text{sca}}(\theta, \phi) \end{aligned}, \quad (11)$$

where $\mathcal{E}_{\text{sca}}(\theta, \phi)$ denotes the angle-dependent part of the far-field electric vector. Eventually, with the orthogonality relations:

$$\begin{aligned} \int_{\Omega} [\mathbf{B}_{lm}(\theta, \phi) \cdot \mathbf{C}_{lm}^*(\theta, \phi)] d\Omega &= 0, \\ \int_{\Omega} [\mathbf{B}_{lm}(\theta, \phi) \cdot \mathbf{B}_{l'm'}^*(\theta, \phi)] d\Omega &= \int_{\Omega} [\mathbf{C}_{lm}(\theta, \phi) \cdot \mathbf{C}_{l'm'}^*(\theta, \phi)] d\Omega = \frac{1}{(D_{lm})^2} \delta_{ll'} \delta_{mm'}. \end{aligned} \quad (12)$$

we have

$$a_M(l, m) = \frac{iD_{lm}}{(-i)^l} \int_{\Omega} [\mathbf{C}_{lm}^*(\theta, \phi) \cdot \mathcal{E}_{\text{sca}}(\theta, \phi)] d\Omega, \quad (13)$$

$$a_E(l, m) = \frac{iD_{lm}}{(-i)^l} \int_{\Omega} [\mathbf{B}_{lm}^*(\theta, \phi) \cdot \mathcal{E}_{\text{sca}}(\theta, \phi)] d\Omega,$$

for the magnetic and electric scattering coefficients. Eq. (13) offers a universal method to evaluate the scattering coefficients with the field vector in the far zone.

To test the validity of the method, we calculate the scattering coefficients for the nanorod with $L = 350$ nm, where $\mathcal{E}_{\text{sca}}(\theta, \phi)$ is obtained from the FDTD simulation (Fig. S20a). The coefficients for $l \geq 3$ are negligible, thus the highest mode supported by this particle is $l = 2$ (i.e., quadrupole). The total radiation intensity distribution (see Fig. S20b) is calculated via these coefficients and Eq. (11), which agrees well with our FDTD simulation results (Fig. 2d, second row). Note that in the calculation of the total intensity $|\mathcal{E}_{\text{sca}}(\theta, \phi)|^2$, there are product terms of the coefficients with different order or type, which represent the contribution from multipolar interference. For example, the summation of the terms related to $a_E(1, m)a_E^*(2, m)$ denotes the ED-EQ contribution. These interference components can be either positive or negative; they do not contribute to the scattering cross-section due to the orthogonal relations (Eq. (12)), but may result in the asymmetric scattering and hence, the recoiling optical force.

Figure S20. Multipole expansion and reconstruction of the scattered field. (a) Calculated scattering coefficients for the cylinder with $L = 350$ nm and $d = 80$ nm. (b) Reconstructed radiation pattern in the x - y plane.

Comment 2: *To support the switchable lateral optical force in Figure 4, the authors should measure the scattering pattern from the nanowire at different linear polarization angle of an incident light.*

Reply 2: We have measured the scattering pattern from a single Ag nanowire under different polarization angles.

Added in main text (Page 13): We measured the scattering pattern from a single Ag nanowire under different polarization angles (Fig. S13), and the measurements show an asymmetry in scattering which supports the recoil-based origin of the LOF.

Supplementary Figure 13. Tuning of far-field scattering by polarization modulation. (a) The experimental setup. A Bertrand lens is removed or inserted before the microscope's image plane, which makes it possible to switch between real- and Fourier space imaging. (b) Measured full Fourier image of a single Ag nanowire under different polarization angles. Asymmetric scattering pattern is observed when the polarization angle is ± 30 degrees. The length of the nanowire is $\sim 3.9 \mu\text{m}$. The FPOL is orientated along x -axis. The numerical aperture of the objective is 1.2.

Comment 3: *What is the temperature rise on the Ag nanowires in Figure 4? Given the strong photothermal response of Ag nanowires, it is possible that an appreciable temperature rise could occur, which may give rise to thermophoretic motion in the fluid. The authors should compute the temperature field distribution on the nanowire and the surrounding fluid medium for the laser intensity used in the experiment. Is there any thermophoretic force on the nanowire? The sign of the thermophoretic force does not change depending on the incident polarization angle?*

Reply 3: We have evaluated the laser-induced thermal effects in our system. Our simulations show that when using a laser intensity of $4 \text{ mW}/\mu\text{m}^2$, the maximum temperature increase is $\sim 21 \text{ K}$ under a polarization angle of ± 30 degrees. The thermophoretic force profile is calculated from temperature gradient, which is symmetric with respect to the long axis of the nanowire. The sign of the thermophoretic force depends on the sign of thermo-diffusion coefficient, which is usually positive for metal particles. As a result, changing the polarization does not affect the sign of thermophoretic force.

Added in main text (Page 13): To rule out alternative explanation to the lateral force such as thermophoretic motion in the fluid, we also calculated the temperature field distribution on the nanowire and the surrounding fluid medium (Fig. S15). Our simulations show that when using a laser intensity of $4 \text{ mW}/\mu\text{m}^2$, the maximum temperature increase is $\sim 21 \text{ K}$. Note that the thermophoretic force, which is symmetric with respect to the long axis of the nanowire, does not contribute to the lateral force. Moreover, the sign of the thermophoretic force depends on the sign of thermo-diffusion coefficient, which is usually positive for metal particles⁵⁰. As a result, changing the polarization does not affect the sign of the thermophoretic force. Thus, the observed effect cannot be due to thermophoretic forces, and the optical force is responsible.

Supplementary Figure 15. Photothermal effect of a single Ag nanowire. (a) Calculated intensity distribution of the electric field with two different polarization angles. (b) Calculated temperature field distribution on the nanowire and the surrounding medium (x - y plane) corresponding to (a). (c) Simulated temperature gradient corresponding to (b). The white arrows represent the thermophoretic force vectors. The laser intensity is $4 \text{ mW}/\mu\text{m}^2$.

Comment 4: *Why do not you submit the real-time movies that shows the lateral force switched by the incident polarization angle in Figure 4d? They would be very insightful for the readers.*

Reply 4: We have submitted the experimental videos in the revised manuscript.

Reviewer 3

Referee report on manuscript NCOMMS-23-12101, "Creating tunable optical forces..."

This paper deals with the lateral optical forces on metallic nanowires (Ag) of different aspect ratios. The work is both theoretical and experimental.

The authors show through calculations that a silver nanowire illuminated by a plane wave, hence an optical field that is homogeneous in space, still can produce a force sideways, in a direction perpendicular to the propagation direction of the light. For this to occur some conditions must be met that produce a lowering of the symmetry. The wave must hit the nanowire at an angle of incidence that differs from zero and moreover, the incident light, while linearly polarized, must consist of a mixture of p and s polarization. The sideways, lateral, force is the result of the recoil forces that occur when photons are predominantly scattered in one of the lateral directions due to an interference between the magnetic and electric multipoles of the nanoparticle. The authors also show experimental results that support the theoretical results.

I find the basic results and conclusions presented in the manuscript to be highly believable and scientifically sound. I also think that the discussion and results presented in the manuscript will inspire further research in this field. The paper contains a solid list of very relevant references to the literature concerning the optical forces.

I think the authors have used methods that are appropriate in conducting the research. The theoretical work is built on a combination of a simple dipole model that clearly illustrates what is happening, and detailed numerical calculations using the FDTD method. I myself am not sufficiently knowledgeable about the experimental methods to assess the experimental methods used in detail.

We thank the reviewer for his/her positive evaluation and for recognizing that our work will inspire further research in this field. We have added more experimental details in the revised manuscript involving how to design the flattop phase-gradient optical line trap to manipulate single nanowires.

Comment 1: *I can identify one apparent weakness in terms of the applicability of this work. It has to do with the fact that the authors consider a very specific configuration of the nanowire in which it is always oriented with its long axis in the same plane as the wave vector of the incident light (the plane of incidence). It is not unlikely that fluctuations and/or optical torques would turn the nanowire away from that position and the manuscript does not address this question at any length. In Figure 4 concerning the optical sorting experiment that the authors have carried out one can see that the nanowires to begin with stay oriented along the optical line but that they eventually leave that line (which happens by design), and then turn away from the original orientation. Hence, I think the manuscript would be strengthened by a discussion concerning the robustness of the orientational alignment of the nanowire.*

Reply 1: We thank the reviewer for raising this important issue. The first reviewer also had similar concerns. We have added some comments on the nanorods/nanowires rotating with linear polarization (Please see our reply 4 to the first reviewer). Similarly, the nanowire also rotates in a linearly polarized optical line trap. In Figure 4, the use of an optical line trap introduces a competition between attractive intensity gradient force (which could be used to keep the orientation of the nanowire) and repulsive lateral force in a single Ag nanowire. In our experiments, we find that when the polarization direction changes from 90° to 30° , the orientation of the nanowire is still parallel with the optical line or tuned depending on the magnitude of the phase gradient. The nanowire rotates and moves to a specific direction (+y or -y) due to the dominant lateral optical force in a well design phase gradient optical line trap. Moreover, the applicability of this work can be extended to the design and fabrication of a new type of micro/nanomotors that will not have the rotation issue (Please see our reply 7 to the first reviewer).

Revised in main text (Pages 12-13): In our experiments, we observed that when the polarization direction changes to $\pm 30^\circ$, the orientation of the nanowire either remains parallel to the optical line or can be tuned depending on the magnitude of the phase gradient. When the polarization angle is -30° , the nanowire stably moves several micrometers along the optical line, then it rotates and is released to a specific direction (e.g., +y) due to the dominant lateral optical force (e.g., positive LOF) within a carefully designed phase gradient optical line trap.

(see Fig. S12 for the measured histogram of the orientation of the nanowire in the FPOL).

Supplementary Figure 12. Measured histogram of the orientation of the nanowire in the FPOL. The polarization direction is perpendicular to the orientation of the nanowire.

Comment 2: *I find the paper to be easy to read for the most part, but I have some suggestions for improvements, primarily concerning the figures. The results for the lateral force as a function of incidence angle in Fig. 1d (the right panel) shows some irregular oscillations. What is the reason for this? In the case of the long wire I can see that there are computational challenges involved, but I do not see why that would also be the case for the short wire.*

Reply 2: For a long wire, multipolar interference also plays an important role in creating LOF (Please see our reply 1 to the second reviewer), which leads to the oscillation of LOF shown in Fig. 1d. In fact, this oscillation is clearly presented in Fig. 3a, where the increment of angle of incidence is 1 degree. The irregular oscillation of LOF applied on short rod is due to FDTD numerical error. We have recalculated the LOF with a finer mesh and revised Fig. 1d.

Comment 3: *In figure 1 a and 1 b, I lack a clear explanation of what the red and blue color mean in the radiation lobes there. I think the caption needs to be more specific there.*

Reply 3: The red and blue color indicates optical scattering along +y and -y directions, respectively. The two colors make it clear which direction dominates overall. We have added a note in the Figure caption.

Comment 4: *The same also goes for Fig. 2c and 2d. Again I do not fully understand what the colors mean. Figure 2c also contains a mistake in that the angle of incidence there is 2 degrees (as the caption says) rather than 0 degrees (as the actual figure says). Finally, concerning Fig 2e the authors talk about a logarithmic scale but it is unclear whether that has been used in producing Fig. 2e.*

Reply 4: We have revised Fig. 2c and 2d. The original Fig. 2e was plotted with logarithmic scale. We have further explored the physics underneath of the LOF, which is not limited to the dipolar landscape. In the revised manuscript, Fig. 2e is replaced by the data of multipolar interplay analysis of a Ag cylinder with a length of 350 nm (Please see our reply 1 to the second reviewer).

Comment 5: *On line 175 and 176 the authors state that the radiation pattern for a short wire (100 nm) shows an "abnormal" dumbbell shape. I would say that that shape is quite normal for dipole radiation.*

Reply 5: Thank you for pointing out this typo. We have changed the “abnormal” to “normal” in the revised manuscript.

Comment 6: *To summarize, I think the paper is scientifically sound and very interesting but in need of a more detailed discussion and explanations on some points.*

Reply 6: We thank the reviewer for his/her positive evaluation. We have made revisions according to the reviewers’ suggestions and comments.

REVIEWER COMMENTS

Reviewer #1 (Remarks to the Author):

The authors have satisfactorily addressed all my concerns and significantly improved their manuscript. I recommend its publication.

Reviewer #2 (Remarks to the Author):

The manuscript has been updated well. However, further major revisions are needed as follows:

In the revised manuscript, the authors have changed the claim about the underlying physics of the lateral force in the revised manuscript from magnetoelectric interplay to multipolar interplay. For example, in the case of a silver nanowire with 350 nm length, the lateral force mainly results from the interference between electric dipole and electric quadrupole, as shown in Fig. 2e. However, the authors still theoretically explain the lateral force based on the magnetoelectric interplay. The magnetoelectric interplay is not the main contribution to the lateral force experimentally demonstrated in Figure 4. In fact, the present theoretical description based on the magnetoelectric interplay is limited to a silver nanowire with 100 nm length. Thus, the present theoretical explanation confuses the reader. I recommend that the authors change the theoretical description of the lateral force from magnetoelectric interplay model to multipolar interplay model, including ED-EQ and EQ-EQ. Otherwise, experimental evidence of the lateral force due to the magnetoelectric interplay should be presented using a silver nanowire with 100 nm length.

Reviewer #3 (Remarks to the Author):

Referee report on revised manuscript (NCOMMS-23-12101A) "Creating tunable lateral optical forces through multipolar interplay in single nanowires":

I have read the authors response to the comments made by me and the other reviewers. I think the authors have handled the recommendations in a good way and I maintain my overall positive opinion on the paper. Hence I recommend that the work is published in Nature Communications.

Response

We thank the Editor and Reviewers for the valuable time. The thoughtful comments provided by the second reviewer are answered point-by-point below, which have significantly strengthened our manuscript (NCOMMS-23-12101A). For convenience, we uploaded a second copy of the manuscript as Supplementary Material, where all revisions are highlighted in red.

Reviewer 1

The authors have satisfactorily addressed all my concerns and significantly improved their manuscript. I recommend its publication.

Reply: We thank the reviewers for recommending the publication of our revised manuscript!

Reviewer 2

The manuscript has been updated well. However, further major revisions are needed as follows: In the revised manuscript, the authors have changed the claim about the underlying physics of the lateral force in the revised manuscript from magnetoelectric interplay to multipolar interplay. For example, in the case of a silver nanowire with 350 nm length, the lateral force mainly results from the interference between electric dipole and electric quadrupole, as shown in Fig. 2e. However, the authors still theoretically explain the lateral force based on the magnetoelectric interplay. The magnetoelectric interplay is not the main contribution to the lateral force experimentally demonstrated in Figure 4. In fact, the present theoretical description based on the magnetoelectric interplay is limited to a silver nanowire with 100 nm length. Thus, the present theoretical explanation confuses the reader. I recommend that the authors change the theoretical description of the lateral force from magnetoelectric interplay model to multipolar interplay model, including ED-EQ and EQ-EO. Otherwise, experimental evidence of the lateral force due to the magnetoelectric interplay should be presented using a silver nanowire with 100 nm length.

Reply: We thank the reviewer for his/her suggestions. We agree with the reviewer that, with the increase of wire length, the main contribution to the lateral force will change from the interplay between electric dipole and magnetic dipole to those higher multipoles. To avoid confusion, we have revised the introduction, by emphasizing the higher multipole effects on the long nanowires, and added a multipolar interplay model in the revised manuscript. While this model does not indicate the LOF directly, it showcases all possible multipolar interactions related to the recoil effects, which is origin of the LOF. And in our subsequent multipolar expansion analysis, we have verified, by observing the light scattering behaviors, that such interactions do contribute to the LOF.

On the other hand, we realize that the dipole model is a commonly used method in the field of optical forces and manipulation, although the objects are sometimes not so small to be approximated as dipoles. For example, in ref. [Nature communications, 2014, 5(1): 3307], the dipole model is employed to show analytically the LOF on a gold helix, which is non-dipolar because its size is comparable with the excitation wavelength. For these reasons, we also retain our analytical results based on the dipole model, because they are simply, clear, and reveal the reiling nature of the LOF, as appreciated by the other two reviewers.

We would like to thank again for the referee's careful reading of our manuscript and their constructive suggestions. We honestly hope that with these changes the revised version provides clarity.

Revised in the introduction (Page 2): Through computational analysis, we find that when the nanowire is short (i.e., can be described by dipoles), the interference between electric dipole and magnetic dipole is the main contribution to the LOF (Fig. 1b). For a longer nanowire, the dominant contributor will switch to the interference between higher multipoles such that the LOF applies to nanowires with various lengths.

Revised in main text (Page 5): Therefore, the LOF will be induced as long as the interference term $\text{Re}(\mathbf{p}^* \times \mathbf{m})$ possesses the lateral component, which is given by $\text{Re}(p_z^* m_x - p_x^* m_z)$. It is evident that the LOF is zero at s -polarization (where $p_x = p_z = 0$) and p -polarization ($m_x = m_z = 0$); the vanishing LOF can also be understood by the radiation symmetry, as exemplified in Fig. 1b-I, II and Fig. 1b-III, IV for s - and p -polarization, respectively. Only when p_z and m_x (or p_x and m_z) coexist could the LOF be produced (Fig. 1b-V, VI), but it also requires the breaking of electric-magnetic symmetry, i.e., $p_x^*/p_z^* \neq m_x/m_z$. These essential prerequisites are easily satisfied in an elongated nanoparticle (e.g., a nanocylinder), whose longitudinal and transverse modes can be excited simultaneously under the illumination of an obliquely incident plane wave.

Added in main text (Page 6): Using the method of Cartesian multipole expansion, one may write the recoil force related to all possible multipoles as^{22,38}

$$\mathbf{F}_{\text{rec}} = \sum_{l=1}^{\infty} \left[\mathbf{F}^{\text{x}(l)} + \mathbf{F}^{\text{e}(l)} + \mathbf{F}^{\text{m}(l)} \right] \quad (6)$$

$$\mathbf{F}^{\text{x}(l)} = \frac{1}{4\pi\epsilon c} \frac{k^{2l+2}}{l!(2l+1)!!} \text{Re} \left[\tilde{\mathbb{O}}_{\text{elec}}^{(l)} \begin{matrix} (l-1) \\ \ddots \end{matrix} \tilde{\mathbb{O}}_{\text{mag}}^{(l)*} \right] \begin{matrix} (2) \\ \ddots \end{matrix} \tilde{\epsilon} \quad (7)$$

$$\mathbf{F}^{\text{e}(l)} = -\frac{1}{4\pi\epsilon} \frac{(l+2)k^{2l+3}}{(l+1)!(2l+3)!!} \text{Im} \left[\tilde{\mathbb{O}}_{\text{elec}}^{(l)*} \begin{matrix} (l) \\ \ddots \end{matrix} \tilde{\mathbb{O}}_{\text{elec}}^{(l+1)} \right] \quad (8)$$

$$\mathbf{F}^{\text{m}(l)} = -\frac{\mu}{4\pi} \frac{(l+2)k^{2l+3}}{(l+1)!(2l+3)!!} \text{Im} \left[\tilde{\mathbb{O}}_{\text{mag}}^{(l)*} \begin{matrix} (l) \\ \ddots \end{matrix} \tilde{\mathbb{O}}_{\text{mag}}^{(l+1)} \right]. \quad (9)$$

where $\tilde{\epsilon}$ is the Levi-Civita tensor, $\mathbb{O}_{\text{elec}}^{(l)}$ denotes the electric 2^l -pole moment, and $\mathbb{O}_{\text{mag}}^{(l)}$ denotes the magnetic 2^l -pole moment; for $l = 1$, $\mathbb{O}_{\text{elec}}^{(1)} = \mathbf{p}$ and $\mathbb{O}_{\text{mag}}^{(1)} = \mathbf{m}$ so that Eq. (7) reduces to \mathbf{F}^{em} . Eqs. (6)-(9) indicates that the recoil force is not limited to the dipoles, but is also found in higher multipoles, including their hybrid magnetoelectric interaction [Eq. (7)] and the interaction between purely electric (or magnetic) modes [Eqs. (8) and (9)]. These multipolar recoil forces can have the lateral component in our system. We shall show below that the higher multipoles indeed contribute to the LOFs on longer nanocylinders, by observing light scattering behaviors.

Reviewer 3

Referee report on revised manuscript (NCOMMS-23-12101A) "Creating tunable lateral optical forces through multipolar interplay in single nanowires":

I have read the authors response to the comments made by me and the other reviewers. I think the authors have handled the recommendations in a good way and I maintain my overall positive opinion on the paper. Hence I recommend that the work is published in Nature Communications.

Reply: We thank the reviewers for recommending the publication of our revised manuscript!

REVIEWERS' COMMENTS

Reviewer #2 (Remarks to the Author):

The authors have fully addressed my concerns. I recommend the final version to be published in Nature Communications.

Reviewers' Comments:

Reviewer #2 (Remarks to the Author):

The authors have fully addressed my concerns. I recommend the final version to be published in Nature Communications.

Response

We thank the reviewer for reviewing and recommending the publication of our revised manuscript!